# Backlight and dim space object detection based on a novel event camera

Xiaoli Zhou[1,2] and Chao Bei[2]

[1] Graduate School, The Second Research Academy of CASIC, Beijing, China
[2] CASIC Space Engineering Development Co., Ltd., Beijing, China

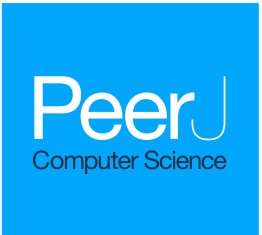

## ABSTRACT

**Background:** For space object detection tasks, conventional optical cameras face various application challenges, including backlight issues and dim light conditions. As a novel optical camera, the event camera has the advantages of high temporal resolution and high dynamic range due to asynchronous output characteristics, which provides a new solution to the above challenges. However, the asynchronous output characteristic of event cameras makes them incompatible with conventional object detection methods designed for frame images.

**Methods:** Asynchronous convolutional memory network (ACMNet) for processing event camera data is proposed to solve the problem of backlight and dim space object detection. The key idea of ACMNet is to first characterize the asynchronous event streams with the Event Spike Tensor (EST) voxel grid through the exponential kernel function, then extract spatial features using a feed-forward feature extraction network, and aggregate temporal features using a proposed convolutional spatiotemporal memory module ConvLSTM, and finally, the end-to-end object detection using continuous event streams is realized.

**Results:** Comparison experiments among ACMNet and classical object detection methods are carried out on Event_DVS_space7, which is a large-scale space synthetic event dataset based on event cameras. The results show that the performance of ACMNet is superior to the others, and the mAP is improved by 12.7% while maintaining the processing speed. Moreover, event cameras still have a good performance in backlight and dim light conditions where conventional optical cameras fail. This research offers a novel possibility for detection under intricate lighting and motion conditions, emphasizing the superior benefits of event cameras in the realm of space object detection.

## INTRODUCTION

The number of spacecraft, debris and objects in space is rapidly increasing, escalating collision risks for human space activities. A space object detection method is needed to coordinate satellite operations and avoid collision risks by sensing space situational awareness (*Sun & Yu, 2019*). Optical detection provides an efficient and cost-effective method for space situational awareness missions, but conventional optical cameras are limited in challenging lighting conditions and suffer from motion blur when capturing

Corresponding author
Xiaoli Zhou,
zhouxiaoli0930@163.com

high-speed moving objects. Event cameras are expected to supplement the shortcomings of conventional optical cameras by providing supplementary sensing information. Event cameras simulate the retina of natural organisms, perceive light changes based on asynchronous events and output asynchronous event streams (*Delbruck, 2016*). They have high temporal resolution, high dynamic range, less motion blur, sub-millisecond latency, and low power consumption. Therefore, event cameras are natural motion detectors with advantages in moving object detection.

Some preliminary work has demonstrated the effectiveness of event cameras in space object detection. *Cohen et al. (2019)* showed they can observe resident space objects from LEO to GEO during both day and night. *Afshar et al. (2020)* compared event cameras with conventional optical cameras, showing higher temporal resolution, lower data redundancy, and no blur in observing moving objects. *Roffe et al. (2021)* tested event camera feasibility in the high-radiation environment of low orbit through physical simulation, proving stable operation despite neutron radiation. *Zhou et al. (2023)* proposed an event stream denoising method, solving the problem of large noise output from event cameras. This ensures accuracy and responsiveness in space scenes with low signal-to-noise ratio, laying a foundation for space object detection.

Utilizing event cameras to address the challenges of backlight and dim space object detection represents a novel research direction, which presents two main challenges:

Firstly, event cameras have an asynchronous output characteristic, making them incompatible with classical vision methods for frame images. New methods are needed, with model-based and spiking neural network (SNN) options being popular. Model-based methods (*Lagorce et al., 2017*; *Orchard et al., 2015*) provide low-latency prediction but require significant computation and focus more on local information. SNN is data-driven but lacks robust learning rules (*Gehrig et al., 2020c*; *Xiao et al., 2021*). A viable option is to convert events through batch processing into image-like representations, using methods like convolutional neural network (CNN) (*Gehrig et al., 2019*; *Messikommer et al., 2020*; *Jiang et al., 2019*; *Redmon et al., 2016*; *Zhang et al., 2021*; *Shariff et al., 2022*). Although this is effective for event-based object detection, it abandons the asynchronous and spatiotemporal correlation characteristics of space objects, leading to issues with isolated event image detection and redundant computation. Therefore, other data representations bring new challenges to method development.

In addition, efficient end-to-end learning methods require large event datasets for training, but space event datasets are difficult to obtain due to novelty of event cameras. Event simulators are a viable alternative, but simulation scenes lack realism (*Rebecq, Gehrig & Scaramuzza, 2018*). *Ralph et al. (2023)* built two mobile observatories in Australia, equipped with two event cameras DVS346 and Gen4 HD, and made progress by observing dense star fields. However, these datasets still lack real space scenes. The lack of large-scale space event datasets hinders development of spaceborne event cameras and limits evaluation of method performance and generalization.

Thus, we propose an asynchronous convolutional memory network (ACMNet) for backlight and dim space object detection with event cameras. The network uses asynchronous event streams as input and combines the feed-forward feature extraction

network CSP-DarkNet with the convolutional spatio-temporal memory module ConvLSTM. Additionally, we build a large-scale space synthetic event dataset from existing real space videos using conventional cameras. The dataset contains many challenging scenes and enables development and evaluation of event-based space object detection methods. The main contributions of this article are as follows:

(1) We proposed a new backlight and dim space object detection network, ACMNet, using event cameras. ACMNet inputs continuous asynchronous event streams, converting them to EST voxel grid representation *via* an exponential kernel function. It uses the CSP-DarkNet for feature extraction. A convolutional spatio-temporal memory module, ConvLSTM, is used to memorize temporal information, aiding in effective space object detection.

(2) We build a new synthetic event dataset, Event_DVS_space7, using a video-to-event generation model and space video data. The dataset features a large number of scenes where conventional optical cameras have limitations, such as backlighting and dim lighting. This dataset enables the development and evaluation of event-based space object detection methods.

(3) We conduct comprehensive experiments to verify the effectiveness of our ACMNet which consistently outperforms the classical object detection methods, and especially has outstanding performance in challenging space scenes.

To the best of our knowledge, this is the first study to investigate such an event-based space object detector in an end-to-end manner. We believe that our ACMNet acts as a bridge between event cameras and practical applications that rely on high-accuracy space object detection.

## RELATED WORK

### Event cameras

An event camera is a new type of neuromorphic visual sensor inspired by the biological visual system, and differs from frame-based visual sensor in the principle of operation. The exposure time of a frame camera is fixed, and the exposure is repeated even if there's no change in illumination on a pixel. In contrast, each pixel of an event camera independently detects changes in illumination, generating asynchronous event data, which includes timestamps, pixel addresses, and the polarity of illumination change.

Event cameras have independent asynchronous pixels that report triggered local illumination changes. The circuit for a single pixel is shown in Fig. 1A. For each pixel, whenever the illumination change reaches a set threshold, it outputs an asynchronous event, as shown in Fig. 1B, where $V$ represents the logarithm of the illumination intensity and t represents the time. If the log-compressed light of the pixel increases by a fixed amount, the pixel sends an ON event asynchronously. If the log-compressed light of the pixel increases by a fixed amount, the pixel sends an OFF event asynchronously. This event-based asynchronous data format is called address event representation (AER) (*Delbruck et al., 2010*) and is used to simulate the transmission of neural signals in biological vision systems, as shown in Fig. 1A. With this representation, information is

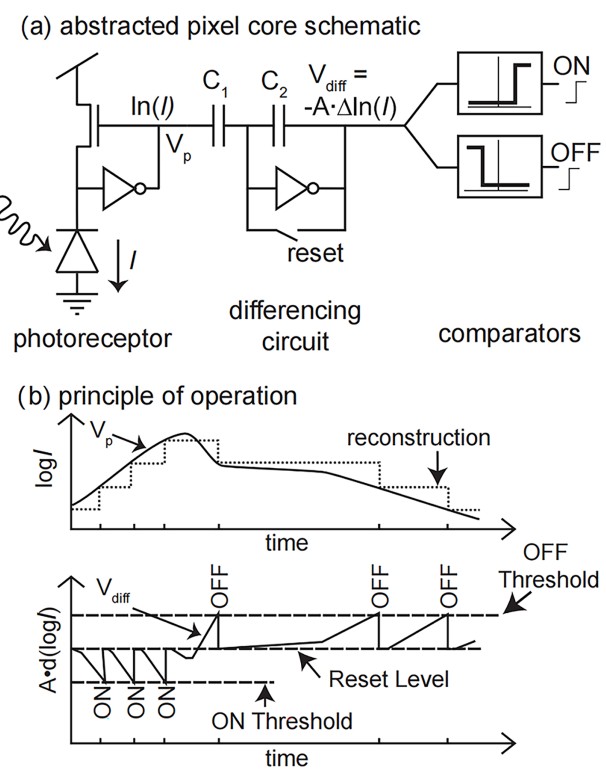

**Figure 1** **A schematic representation of the event camera pixels.** (A) Core schematic. (B) Principle of operation.

continuously transmitted and processed, and the communication bandwidth is only occupied by the pixel that triggers the event.

## Event-based object detection

Recent advancements in event-based object detectors have addressed deficiencies of traditional optical cameras in complex visual scenarios. Current methodologies can be categorized into event images (*Chen, 2018*; *Cannici et al., 2019*; *Iacono, Weber & Glover, 2018*; *Li et al., 2019*) and event volumes (*Hu, Delbruck & Liu, 2020*; *Perot et al., 2020*). Event images convert asynchronous events into binary or grayscale images for use in traditional frame-based detectors. Event volumes integrate event data into continuous volumetric format. Both types generate 2D, image-like representations compatible with frame-based systems, but struggle to fully exploit spatio-temporal characteristics of asynchronous events.

A critical review of existing methods (*Messikommer et al., 2020*; *Iacono, Weber & Glover, 2018*; *Li et al., 2019*; *Hu, Delbruck & Liu, 2020*; *Perot et al., 2020*) indicates that these approaches predominantly utilize a simplistic strategy by deploying a feed-forward, frame-based detection model independently for each event image, which fails to capitalize on the rich temporal cues and motion information. This limitation has been primarily due to the absence of large-scale, continuous event datasets that are amenable to supervised learning applications, before the release of the Gen1 automotive dataset (*Tournemire et al., 2020*) and the 1Mpx detection dataset (*Perot et al., 2020*). Drawing on principles from

video object detection (*Han et al., 2016*; *Chen, Yu & Wu, 2020*; *Zhu & Liu, 2018*), which make use of temporal information across multiple adjacent frames, this article proposes a novel method for event-based space object detection, which distinctively processes asynchronous events and harnesses rich temporal cues through an innovative asynchronous convolutional memory architecture, aiming to substantially improve the utilization of temporal data in event-based detection systems.

## Event representation

A fundamental issue in the emerging field of event cameras is the optimal extraction of spatio-temporal representations from asynchronous events for specific applications (*Gehrig et al., 2019*; *Roy, Jaiswal & Panda, 2019*; *Gallego et al., 2022*). The literature categorizes existing methods into four primary approaches: image-like conversions, handcrafted descriptors, spiking neural networks (SNNs), and deep neural networks (DNNs). Initial approaches involved transforming asynchronous events into image-like planes, either within a fixed temporal window or based on a constant event count (*Maqueda et al., 2018*; *Zhu et al., 2019*). These were employed in various computer vision tasks such as image reconstruction (*Rebecq et al., 2021*), object classification (*Amir et al., 2017*; *Wang et al., 2019*), detection (*Jiang et al., 2019*; *Li et al., 2019*), and tracking (*Gehrig et al., 2020b*). Furthermore, some studies have developed efficient spatiotemporal feature descriptors from asynchronous events, including temporal surfaces (*Lagorce et al., 2017*; *Sironi et al., 2018*) and DART (*Ramesh et al., 2019*). However, these descriptors, often utilized in corner and edge detection, are not only time-intensive to design but also heavily dependent on the dynamics of the moving objects involved. SNNs represent another approach, where end-to-end learning of representations is theoretically possible (*Shrestha & Orchard, 2018*; *Paredes-Valles, Scheper & Croon, 2020*; *Neftci, Mostafa & Zenke, 2019*). Despite this, SNNs have yet to achieve performance on par with DNNs and have predominantly been applied to classification tasks, not extending to numerical regression challenges. In recent developments, there has been a significant push towards creating end-to-end learning frameworks using DNNs. Innovations such as EST (*Gehrig et al., 2019*), Matrix-LSTM (*Messikommer et al., 2020*), and Sparse-Conv (*Cannici et al., 2020*) have been specifically designed to directly process asynchronous events, achieving cutting-edge results. These advances highlight the increasing efficacy and applicability of DNNs in harnessing the full potential of event-based data for complex computational tasks.

## Long short-term memory (LSTM)

Recurrent neural networks (RNNs) (*Yu et al., 2019*) and their variants, including long short-term memory (LSTM) (*Hochreiter & Schmidhuber, 1997*) and gated recurrent units (GRU) (*Cho et al., 2014*), are well-known for handling sequential inputs and modeling long-term dependencies. Recently, these networks have been integrated with convolutional neural networks (CNNs) (*LeCun, Bengio & Hinton, 2015*) to create recurrent convolutional architectures, particularly for video-related tasks (*Zhu & Liu, 2018*; *Cruz & Bernardino, 2019*; *Li, Liu & Wang, 2019*; *Lai et al., 2020*; *Wang et al., 2021*). These architectures aim to leverage both spatial and temporal memory.

Several recurrent convolutional architectures have been developed for video object detection (*Zhu & Liu, 2018*; *Cruz & Bernardino, 2019*). For example, *Cruz & Bernardino (2019)* used convolutional LSTM to capture temporal features in maritime airborne video object detection. Similarly, *Zhu & Liu (2018)* introduced a bottleneck LSTM layer that reduces computational demands compared to traditional LSTM configurations. Despite these advancements, these systems often suffer from high computational complexity, which can be a limiting factor.

To address these challenges, this article introduces a convolutional spatiotemporal memory module ConvLSTM designed to efficiently extract temporal features. This module stacks multiple ConvLSTM layers to construct a robust detection framework suitable for a feed-forward event object detector leveraging asynchronous events. Our architecture aims to diminish computational costs while enhancing performance and preserving the capacity to maintain long-term dependencies effectively.

### Synthetic events

An overview of event modeling using event cameras is provided. Early research by *Kaiser et al. (2016)* developed a basic method of event generation by applying a threshold to image frame differences. Pix2NVS (*Bi & Andreopoulos, 2017*) calculates per-pixel brightness in conventional video frames to generate events. The first accurate event generators are in references (*Mueggler et al., 2017*) and *Li et al. (2018)*, using high frame rate rendering and linear interpolation of intensity signals. *Rebecq, Gehrig & Scaramuzza (2018)* proposed an adaptive sampling mechanism, adjusting to maximum displacement between frames. The generative models in *Rebecq, Gehrig & Scaramuzza (2018)* and *Mueggler et al. (2017)* are underpinned by formalizations established in earlier studies (*Lichtsteiner, Posch & Delbruck, 2008*; *Gallego et al., 2015*, *2017*). These foundations significantly contributed to advancing and understanding event-based modeling, highlighting potential applications.

## PROPOSED METHOD

### Overview of the ACMNet

Due to the particularity of the space scene, the target is usually far away and small, and lacks significant appearance features under extreme lighting conditions. It is difficult to separate the target from the background by appearance features alone, and a large number of motion features are hidden in the time series information, which is easy to be ignored. Therefore, an extraction method combining appearance features and timing features is proposed in this article, and based on this method, an asynchronous convolutional memory network (ACMNet) for spatial object detection is constructed. The network framework is shown in Fig. 2. Taking asynchronous event streams as input, the kernel function is converted into EST voxel grid to retain timing features. After the appearance features are extracted in the feedforward feature extraction network, the BiSA-ConvLSTM module is constructed to memory time series features, and multi-scale feature fusion and regression prediction algorithm are used to output accurate target detection results.

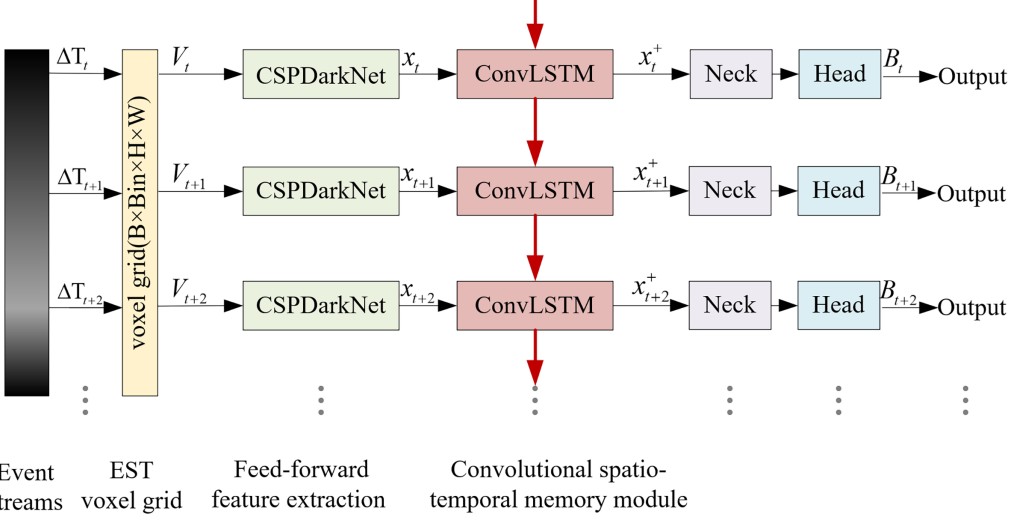

**Figure 2 Overview of the ACMNet framework.**

The algorithm process is as follows:

Input: Space event stream data collected by the event camera.

Output: The detection results (object category and confidence).

(1) Segment the asynchronous event stream into fragments, and represent them using a three-dimensional event field $S$. After convolution with the exponential kernel function $k(u, t)$, the resulting representation is converted to a voxel grid $V$.

(2) Use a forward feature extraction network to extract spatial scale features, obtaining the forward feature tensor $x_t$.

(3) For the voxel grids $V_t$ of the same time batch, the forward feature tensor $x_t$ and the previous time batch memory state $h_{t-1}$ are inputted to the convolutional temporal memory module $\mathcal{L}$, which extracts the temporal sequence features of the target, resulting in the temporal feature tensor $x_t^+$.

(4) Implement multi-scale feature fusion and regression prediction using the Neck and Head modules, outputting the detection results $B_t$.

## EST voxel grid

Under the conditions of backlight and dim light, the space object usually lacks significant appearance features, so it is very important to extract the motion information of the object. These motion information are fused into the temporal information of the event stream, so it is necessary to transform the event stream data into a representation that can effectively retain the timing characteristics. The unique representation of EST voxel grid can effectively preserve the event and its temporal characteristics with high time resolution, which is especially critical for detecting the backlight and faint targets with weak appearance characteristics. Therefore, the characterization method of EST voxel grid was adopted in this article.

Event cameras have pixels that are independent and respond to changes in the continuous log brightness signal $L(u, t)$. Event cameras mark a time stamp with a

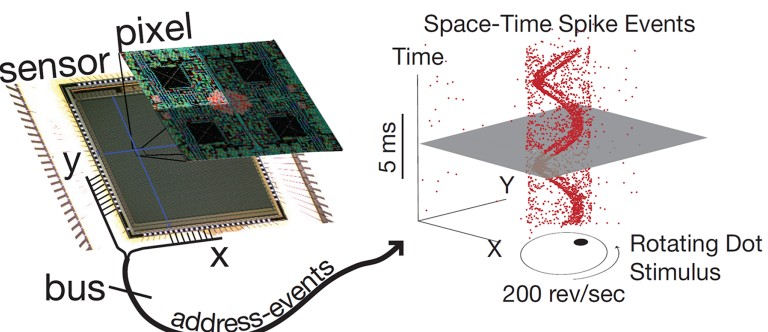

**Figure 3 Event camera schematic diagram.**

microsecond-level resolution and output asynchronous events, as shown in Fig. 3. This example from the dynamic vision sensor shows how a high-speed stimulus generates a sparse, asynchronous digital stream of events which rapidly signify changes in scene reflectance (*Delbruck et al., 2010*).

An event $e_k = (x_k, y_k, p_k, t_k)$ is triggered when the magnitude of the log brightness at the pixel $u_k = (x_k, y_k)^T$ and time $t_k$ has changed by more than a threshold $C$ since the last event at the same pixel.

$$\Delta L(u, t_k) = L(u, t_k) - L(u, t_k - \Delta t_k) \geq p_k C \tag{1}$$

where, $\Delta t_k$ is the time since the last triggered event, $p_k \in \{-1, +1\}$ is the sign of the change, also called the polarity of the event.

In a given temporal interval, the event camera will trigger many events:

$$E = \{e_i\}_{i=1}^N = \{(x_i, y_i, t_i, p_i)\}_{i=1}^N \tag{2}$$

Due to the asynchronous characteristic, events are represented as a set. To use events in combination with a convolutional neural network, it is necessary to convert the event set into a grid-like representation. Events represent point-sets in a four-dimensional manifold spanned by the x and y spatial coordinates, time, and polarity. This point-set can be summarized by the event field, inspired by *Rebecq, Gehrig & Scaramuzza (2018)* and *Gallego et al. (2022)*:

$$S(x, y, t) = \sum_{e_i \in E} p_i \delta(x - x_i, y - y_i) \delta(t - t_i) \tag{3}$$

This representation replaces each event by a Dirac pulse $\delta$ in the space-time manifold. We convolve it with an exponential kernel function $k(u, t)$, and the convolved signal thus becomes:

$$(k * S)(x, y, t) = \sum_{e_i \in E} p_i k(x - x_i, y - y_i, t - t_i)$$

$$k(x, y, t) = \delta(x, y) \frac{1}{\tau} \exp(-t/\tau) \tag{4}$$

After kernel convolutions, a grid representation of events can be realized by sampling the convolved signal at regular intervals:

$$V[x_w, y_h, t_b] = (k * S)(x_w, y_h, t_b) = \sum_{e_i \in E} p_i k(x_w - x_i, y_h - y_i, t_b - t_i) \qquad (5)$$

$[x_w, y_h, t_b]$ is the spatiotemporal coordinates of the EST voxel grid, $x_w \in \{0, 1, \cdots, W - 1\}, y_h \in \{0, 1, \cdots, H - 1\}, t_b \in \{t_0, t_0 + \Delta t, \cdots, t_0 + Bin\Delta t\}$, where $t_0$ is the first timestamp, $\Delta t$ is the bin size, and $Bin$ is the number of temporal bins.

The representation of EST voxel grid effectively retains the high temporal resolution and spatiotemporal characteristic of the event, especially for small and faint objects that lack obvious spatial characteristics, this representation retains the characteristic of the time dimension, which is beneficial to subsequent object detection.

## Feed-forward feature extraction network (CSP-DarkNet)

CSPDarkNet includes a focus structure and a CSP structure. Focus structure is a convolutional neural network layer for feature extraction, which is used as the first convolutional layer in the network to down-sample the input feature map, compress and combine the information in the feature map, to extract a higher-level feature representation, and reduce the amount of computation and parameters. The original feature map is input into a focus structure, sliced, and turned into a $320 \times 320 \times 12$ feature map. The slicing operation is shown in Fig. 4.

The core idea of the CSP structure is that the input feature graph is divided into two parts, one part is processed by a small convolutional network (called a sub-network) and the other part is directly processed in the next layer. The two feature maps are then spliced together as input for the next layer. It can significantly reduce the parameters and computation of the network, and improve the efficiency of feature extraction, to speed up the training and inference of the model.

In this article, CSP-DarkNet is used as the feed-forward feature extraction network to extract object features at different spatial scales. The input of CSP-DarkNet is an image with a size of $640 \times 640 \times 3$. To transfer CSP-DarkNet to the task in this article, the number of input channels is modified to be $W \times H \times bin$, and the value of batch size is $B$, which is set in training and testing to adapt to the EST voxel grid as network input. The output of the network is $x$ a $W \times H \times bin$ dimensional feed-forward feature tensor.

Asynchronous event streams contain rich temporal information, but existing event-based object detectors (*Messikommer et al., 2020*; *Chen, 2018*; *Iacono, Weber & Glover, 2018*; *Li et al., 2019*; *Jiang et al., 2018*) suffer from the loss of rich temporal information and motion information. Since the feature of the object is not obvious during remote observation, it is necessary to continuously observe and fully analyze and extract temporal information. Therefore, we effectively retain temporal information by transferring the representation of the EST voxel grid and propose to add ConvLSTM, a convolutional spatiotemporal memory module, to CSP-DarkNet to further extract temporal features. The feature extraction network of ACMNet consists of a feed-forward feature extraction

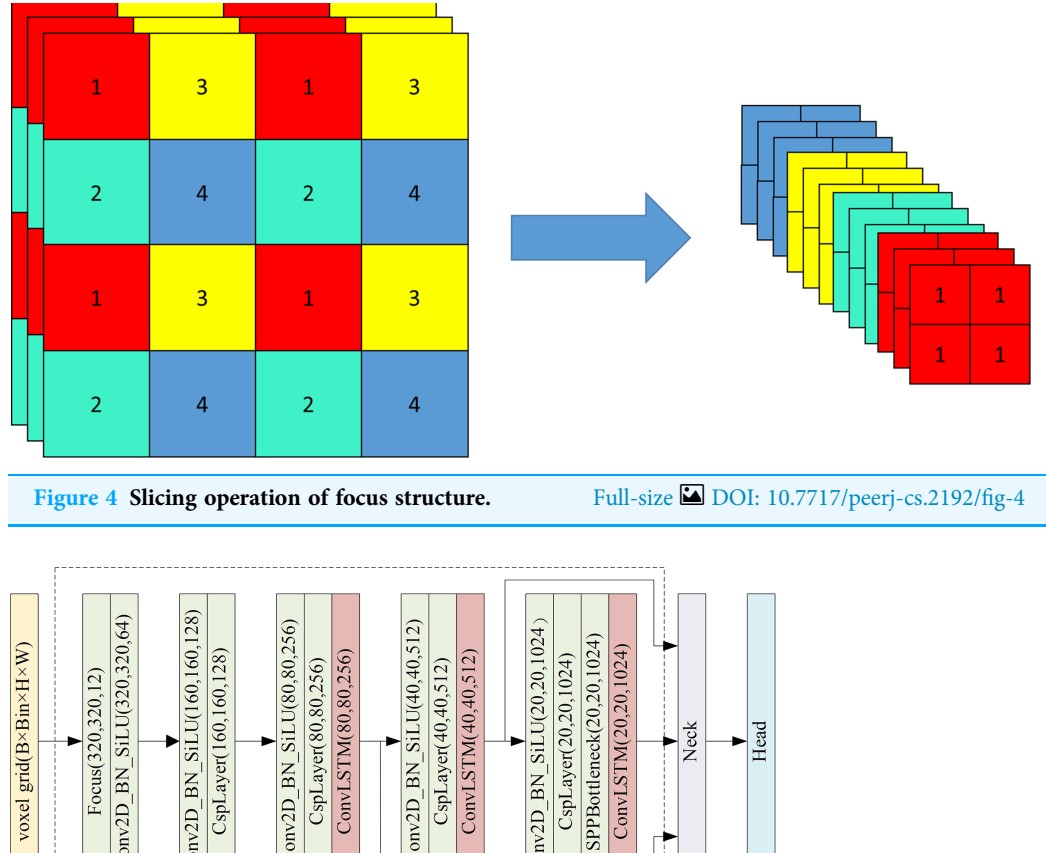

**Figure 4 Slicing operation of focus structure.**

Feature extraction

**Figure 5 Feature extraction network framework.**

network and a convolutional spatio-temporal memory module. Its structure is shown in Fig. 5.

## Convolutional spatio-temporal memory module (ConvLSTM)

Asynchronous event stream is a kind of data format, that contains rich temporal information, especially when dealing with dynamic scenarios. However, the existing event-based object detectors are faced with the problem of loss of temporal and motion information when dealing with such data, especially under complex lighting conditions. When observing long-distance objects, the features are not obvious, this problem is more prominent. To solve the above problems, it is necessary not only to observe the event stream continuously but also to analyze and extract the temporal information deeply. The solution in this article converts the event stream into a voxel grid representation, preserving its temporal information. Then, these temporal features are further extracted in the feature extraction network.

A long short-term memory network (LSTM) is a model used to process time sequence data, which can extract time domain features from arbitrary sequence data

(*Hu, Delbruck & Liu, 2020*). FC-LSTM is an LSTM structure based on a fully connected layer, which better captures the spatial connection relationship by adding a snooping mechanism. However, because it contains too much data, there is a lot of redundancy.

In this article, we extend the conventional FC-LSTM idea to ConvLSTM by replacing the input-to-state and state-to-state calculations in FC-LSTM with convolution operations and forming a detection structure by superimposing multiple ConvLSTM layers to extract temporal correlation features.

ACMNet contains $n$ sub-network structures, expressed as $A = \{a_1, a_2, \cdots, a_n\}$, where the feed-forward feature extraction network CSP-DarkNet includes $m$ sub-network structures, and its output is a feed-forward feature tensor $x_t$, and it can be expressed as:

$$x_t = (a_1 \bullet a_2 \bullet \cdots \bullet a_m) V_t \tag{6}$$

where $\bullet$ denotes a connection between two adjacent layers in the feed-forward feature extraction network CSP-DarkNet.

The CspLayer layer of CSP-DarkNet is connected to ConvLSTM. The internal structure of ConvLSTM is shown in Fig. 6. The equation is as follows:

$$
\begin{aligned}
i_t &= \sigma(W_{xi} * x_t + W_{hi} * h_{t-1} + W_{ci} * c_{t-1} + b_i) \\
f_t &= \sigma(W_{xf} * x_t + W_{hf} h_{t-1} + W_{cf} * c_{t-1} + b_f) \\
c_t &= f_t \circ c_{t-1} + i_t \circ \tanh(W_{xc} * x_t + W_{hc} * h_{t-1} + b_c) \\
o_t &= \sigma(W_{xo} * x_t + W_{ho} * h_{t-1} + W_{co} \circ c_t + b_o) \\
h_t &= o_t \circ \tanh(c_t)
\end{aligned}
\tag{7}
$$

where $*$ denotes convolution operation, $\circ$ denotes Hadamard product.

For a temporal bin $V_t$, ConvLSTM $L$ takes the feed-forward feature tensor $x_t$ and the memory state $h_{t-1}$ from previous temporal bins as the input, then outputs the current state $h_t$ and a spatiotemporal feature tensor $x_t^+$, and it can be described as:

$$(x_t^+, h_t) = L(x_t, h_{t-1}) \tag{8}$$

The last part of ACMNet is the spatiotemporal feature tensor passing through the neck and head modules to output the final detection result, described as:

$$B_t = (a_{m+1} \bullet a_{m+2} \bullet \cdots \bullet a_n) x_t^+ \tag{9}$$

The convolutional spatiotemporal memory module retains the object's temporal information by combining a synchronization grid. LSTM is used to improve the processing ability of the module for time series data, which effectively improves the performance of the object detector in processing asynchronous event streams. Especially under complex illumination and remote observation conditions, this module can significantly improve the recognition ability of object features, thus improving the detection accuracy of the overall network.

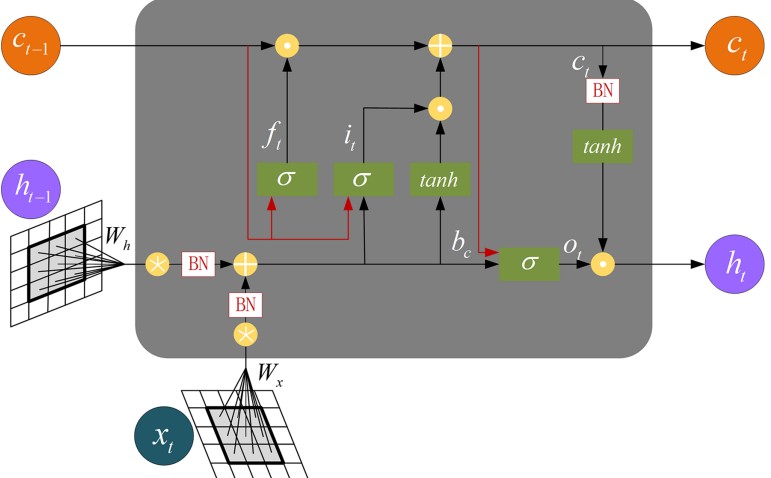

**Figure 6 The internal structure of the ConvLSTM.**

# RESULTS

## Datasets

Since event cameras have not been applied in space so far, there is a lack of data captured by event cameras in space scenes. The space video data captured by spaceborne optical cameras are rich and contain a large amount of space object information. We use the video generation event model to synthesize existing space video data into event data and build a space synthetic event dataset, namely Event_DVS_space7. Space video data are selected to contain a large number of backlight, dim light, and other scenes where the use of conventional optical cameras is limited, to synthesize challenging event data to complete the verification of our proposed method.

### Video to event generation model

Equation (1) defines the event generation model for an ideal event camera (*Gallego et al., 2022, 2017*). We employ a recent frame interpolation method (*Jiang et al., 2018*) to transform low frame rate video into high frame rate video *via* adaptive upsampling. This enhanced video is then utilized to generate events through a generative model using an advanced event camera simulator (ESIM) (*Rebecq, Gehrig & Scaramuzza, 2018*). The number of intermediate samples was determined as previously described in *Jiang et al. (2018)* and *Gehrig et al. (2020a)* using a bidirectional optical flow implementation for internal estimation. Given two consecutive frames $I(t_i)$ and $I(t_{i+1})$ at times $t_i$ and $t_{i+1}$, we generate $K_i$ equally spaced intermediate frames. The value of $K_i$ is selected such that the relative displacement between intermediate frames does not exceed 1 pixel for any pixel:

$$K_i = \max_{\mathbf{u}} \max \left\{ \|F_{i \to i+1}(u)\|, \|F_{i+1 \to i}(u)\| \right\} - 1 \tag{10}$$

where $F_{i \to i+1}(u)$ represents the optical flow from frame $i$ to $j$ at pixel location $u$. This method adaptively upsamples between video frame pairs, resulting in an adaptively upsampled video sequence.

The subsequent step is to generate events from the high frame rate video sequence using Eq. (9). For each pixel, the continuous intensity signal over time is approximated by linearly interpolating between video frames.

Next, synthetic events and original labels are employed to train a network. We use a window of events leading up to the time-stamped ground truth label to train a model to predict it. This method is applicable to general datasets with precisely timestamped images and labels. We leverage existing CNN architectures designed for standard images by converting the asynchronous and sparse event streams into tensor-like representations. We selected the event spike tensor (EST) in "EST Voxel Grid" because it has been shown to outperform other representations in various tasks.

### Space synthetic event dataset

The space synthetic event dataset we built has the following characteristics:

(1) The event generation model adopts the principle of the dynamic vision camera (DVS) model. The time sampling rate is $1.6 \times 10^8$, dynamic range is 120 dB, latency is 1 μs, and the pixel resolution is $640 \times 360$. The synthetic event dataset is called Event_DVS_space7.

(2) The method of building the dataset in this article directly relies on the video data recorded in the real space scenes, so a large number of real space video data, such as backlight and dim light, are selected to generate synthetic event data in the corresponding scene, which provides a basis for the verification of proposed method.

(3) Seven kinds of common space objects are selected to build datasets, including the sun, moon, stars, earth, astronauts, spacecraft, point objects such as small-sized space debris, and long-distance spacecraft. The proportion of various object quantities in Event_DVS_space7 is as follows: sun 7%, moon 4%, stars 6%, earth 31%, astronauts 3%, spacecraft 12%, point objects 37%.

In addition, the building of Event_DVS_space7 also fully considers the working mode of the event camera, the observation distance, and the number of objects and light intensity. Some of the scenes are shown in Fig. 7. Figures 7A and 7B respectively show the frame images and event images of the spacecraft during earth observation and sky survey observation. When the object is close to the detector, it presents surface object characteristics that the outline information can be identified. During long-distance detection, the objects often appear as point objects, including small-sized space debris and long-distance spacecraft, as shown in Figs. 7C and 7D. Figures 7E and 7F show the single object and multiple objects observed by two kinds of cameras respectively. Figures 7G and 7H show the frame image and event image when the light changes from dim light to backlight during the rising of the sun from the horizon, which can highlight the performance of the event camera under different lighting conditions.

We evaluate our proposed method on the Event_DVS_space7 dataset with a resolution of $640 \times 360$ and containing 1,830 samples. Event_DVS_space7 dataset is divided into training, validation, and test sets according to a proportion of 8:1:1.

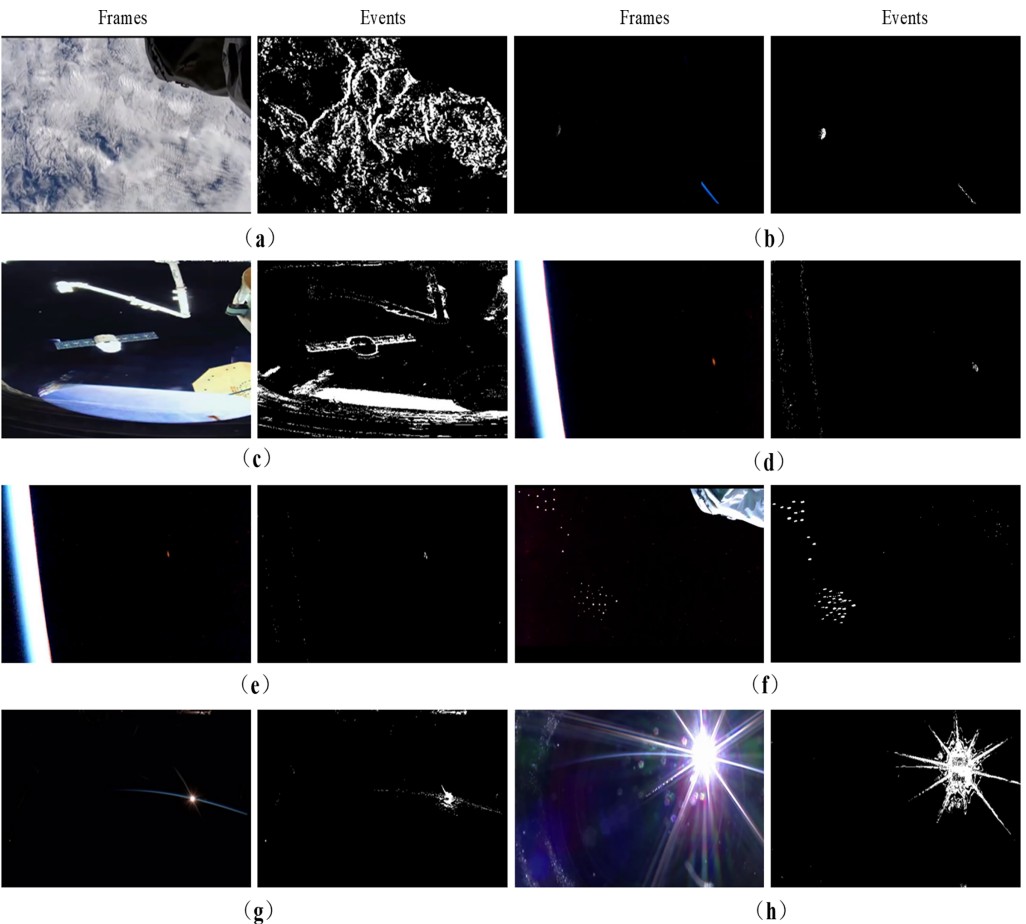

|  Frames | Events | Frames | Events  |

**Figure 7  Partial scenes of Event_DVS_space7 dataset.** (A) Earth observation. (B) Sky survey observation. (C) Surface objects during short-distance detection. (D) Point objects during long-distance detection. (E) Single object. (F) Multiple objects. (G) Dim light. (H) Backlight. Image source: https://doi.org/10.6084/m9.figshare.26005189.v1.               

## Implementation details and metrics

Implementation Details. We train the proposed model using the PyTorch toolbox on the Ubuntu 20.04 LTS operating system. We use the Adam optimizer with a fixed learning rate of 0.0001. We train the model for approximately 500 iterations and 8 batches to obtain the final result. The training is performed on a PC with an i5-13400F processor @4.6 GHz, 16 GB RAM, and an NVIDIA RTX 3060Ti GPU.

Metrics. Three commonly used metrics, average precision (AP), mean average precision (mAP) and runtime, are used to evaluate the performance of ACMNet in this article. Runtime refers to the time consumed from inputting an image to outputting the final result, including pre-processing time (such as image normalization), network front-pass time, and post-processing (such as non-maximum suppression). The calculation formulas of AP and mAP are as follows:

$$AP = \int_0^1 P dR \tag{11}$$

**Table 1 Quantitative comparison with methods in effective experiments.**

| Methods | Representation | mAP | AP | | | | | | | Runtime (ms) |
|---------|---------------|-----|--------|-----------|-----------|------|-----|------|-------|--------------|
| | | | Object | Satellite | Astronaut | Moon | Sun | Star | Earth | |
| Asynet | Sparse-conv | 0.386 | 0.474 | 0.650 | 0.641 | 0.642 | 0.114 | 0.120 | 0.065 | 58.21 |
| Fast R-CNN | Event image | 0.549 | 0.646 | **0.776** | **0.764** | **0.755** | 0.497 | 0.180 | 0.226 | 60.57 |
| SSD | Event image | 0.524 | 0.622 | 0.754 | 0.745 | 0.735 | 0.477 | 0.161 | 0.174 | **53.67** |
| YOLOV3 | Event image | 0.484 | 0.584 | 0.727 | 0.733 | 0.711 | 0.401 | 0.133 | 0.100 | 54.74 |
| YOLOV5 | Event image | 0.514 | 0.602 | 0.751 | 0.741 | 0.73 | 0.474 | 0.145 | 0.158 | 55.84 |
| Gehrig et al. | EST | 0.546 | 0.653 | 0.762 | 0.747 | 0.737 | 0.507 | 0.201 | 0.216 | 64.12 |
| Our ACMNet | EST | **0.641** | **0.681** | 0.765 | 0.757 | 0.748 | **0.548** | **0.579** | **0.412** | 70.14 |

**Note:**
The bold and the underlined signify the best and second-best performance, respectively.

$$\text{mAP} = \frac{\sum\limits_{i=1}^{N} AP_i}{N} \tag{12}$$

where $P$ and $R$ represent precision and recall, respectively; $N$ represents the number of detection categories.

## Effective experiments

Quantitative evaluation, visualization evaluation, and experimental analysis will be highlighted to see why and how our ACMNet works as follows.

### Quantitative evaluation

As shown in Table 1, the bold and the underlined signify the best and second-best performance, respectively. Note that our ACMNet obtains better performance than the existing event-based object detectors (*Gehrig et al., 2019*; *Messikommer et al., 2020*; *Jiang et al., 2019*; *Shariff et al., 2022*; *Ren et al., 2017*; *Liu et al., 2016*). Compared to the benchmark (*Shariff et al., 2022*), our ACMNet gets around 12.7% improvements on the Event_DVS_space7 dataset. At the same time, for the detection of surface objects at a short distance with clearly visible outlines (such as spacecraft, astronauts, sun, and moon), the accuracy of ACMNet is at the best or second-best performance. For long-distance point objects (such as stars), our ACMNet is significantly better than other detectors because of the introduction of two strategies: Firstly, changing the representation method of event streams, that is, using EST voxel grid to allow asynchronous event streams in preserve high-resolution temporal information. In addition, the convolutional spatio-temporal memory module is introduced to extract the temporal features between adjacent event streams to improve the detection performance, which also relatively increases the computational complexity.

### Visualization evaluation

Some representative visualization results on Event_DVS_space7 dataset are illustrated in Fig. 8. Note that, our ACMNet achieves the best performance against state-of-the-art methods including integrating event stream into event images SSD (*Liu et al., 2016*),

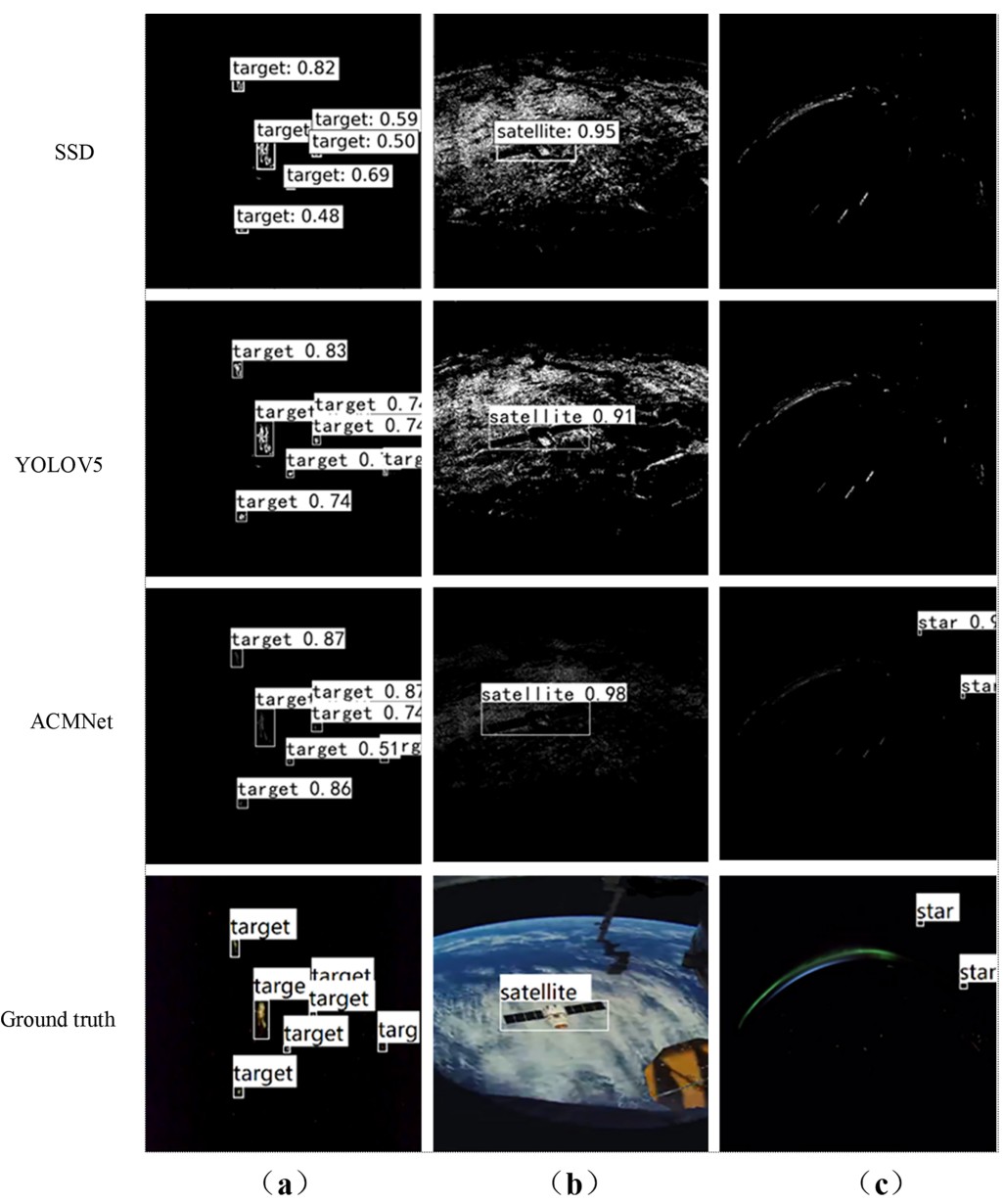

**Figure 8  Visual comparison of different detection methods in effective experiments.** (A) Multi-object scenes of sky survey observation. (B) Short-distance object scenes of earth observation. (C) Tiny object scenes of dim light observation. Image source: https://doi.org/10.6084/m9.figshare.26005189.v1.

YOLOV5 (*Shariff et al., 2022*). Figures 8A–8C are the multi-object scene of sky survey observation, the short-distance object scene of earth observation, and the tiny object scene of dim light observation, respectively. ACMNet shows better detection performance: from the ground truth of Fig. 8A, it can be seen that there are seven objects in the field of view, while the SSD network has one missed detection. As can be seen from Fig. 8B, both SSD and YOLOV5 networks can achieve over 90% accuracy in surface object detection, which is better than point object detection. It can be seen from Figs. 8A–8C that ACMNet shows

**Table 2 The performance of ACMNet components.**

| Methods | VoxelGrid | ConvLSTM | mAP | Runtime (ms) |
|---|---|---|---|---|
| YOLO | | | 0.514 | 55.84 |
| Network I | √ | | 0.556 | 62.69 |
| Network II | | √ | 0.550 | 65.00 |
| Our ACMNet | √ | √ | 0.641 | 70.14 |

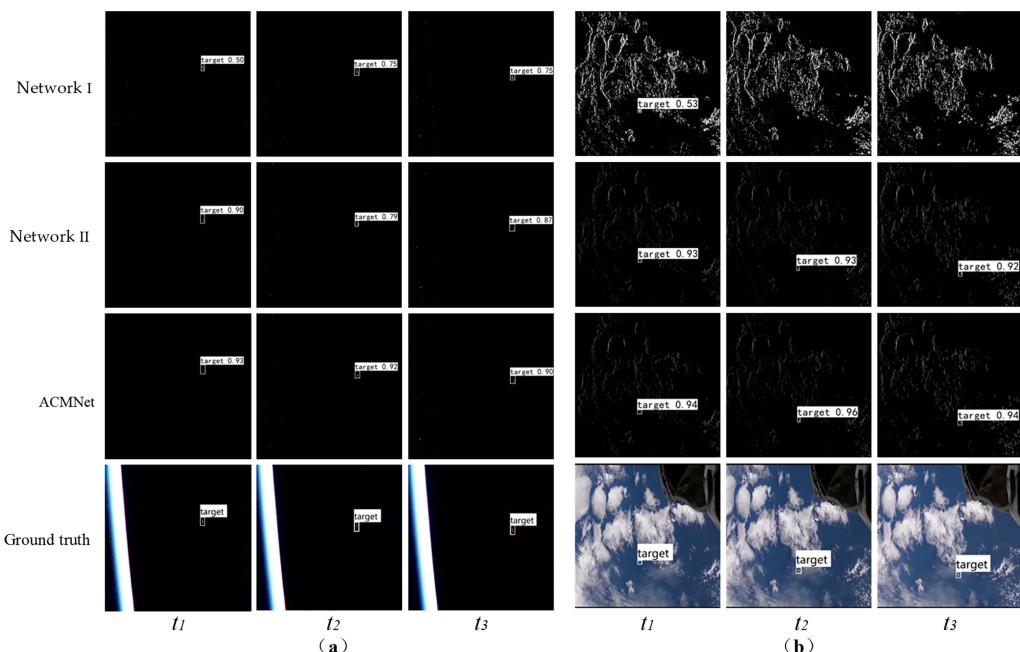

**Figure 9 Visual comparison of detection results in ablation test.** (A) Scenes of earth observation. (B) Scenes of sky survey observation. Image source: https://doi.org/10.6084/m9.figshare.26005189.v1.

better detection performance, especially for the effective detection of tiny objects in dim light conditions, while cannot be captured by SSD and YOLOV5. This is because ACMNet benefits from the representation of events and the convolutional spatio-temporal memory module, which can obtain effective temporal information from event streams. ACMNet is robust and adaptable to object detection with different working modes, detection distances, the number of objects, and light intensity.

## Ablation study

Beyond the effective test, we next conduct ablation tests to take a deep look at the impact of each design choice and parameter settings for our ACMNet, and more details are presented in Table 2 and Fig. 9. As shown in Table 2, we further analyze how much each component to the ultimate performance. Notably, our two baselines, namely network I and II, utilize VoxelGrid and ConvLSTM respectively that, and consistently achieve higher performance than the benchmark (*Shariff et al., 2022*). More precisely, by introducing

**Table 3 Comparison of detection performance between frame images and event images.**

| Architectures | Events | | Frames | |
|---|---|---|---|---|
| | mAP | Runtime | mAP | Runtime |
| Faster-RCNN | 0.639 | 56.24 | 0.549 | 60.57 |
| SSD | 0.604 | 46.35 | 0.524 | 53.67 |
| YOLOV5 | 0.610 | 48.20 | 0.514 | 55.84 |

ConvLSTM module, the detection accuracy of network II increases by 3.6% based on the benchmark, and ACMNet increased by 8.6% compared to network I, indicating that the ConvLSTM module plays a positive role in improving detection accuracy. By introducing the VoxelGrid module, network I obtained 4.2% higher than the benchmark, and ACMNet improved mAP by 9.1% compared to Network II, verifying that the event stream representation method of EST voxel grid is more beneficial to improve the detection performance than the conversion to the event image. In addition, the last row of Table 2 illustrates that the detection speed of our ACMNet, introducing VoxelGrid and ConvLSTM, is almost comparable in contrast to the benchmark (*Shariff et al., 2022*).

Figure 9 presents the results in continuous event stream detection of the baseline network and ACMNet. Among them, Figs. 9A and 9B represent the classical scenes for earth observation and sky survey observation respectively. It can be observed from the Fig. 9 that ACMNet outperforms the baseline network in terms of detection performance. This further validates the effectiveness of these two modules in space object detection. Additionally, the baseline network without the convolutional spatiotemporal memory module (network I) performs better than other object detectors (*Gehrig et al., 2019*; *Messikommer et al., 2020*; *Jiang et al., 2019*; *Shariff et al., 2022*; *Ren et al., 2017*; *Liu et al., 2016*), demonstrating both high detection accuracy and fast detection speed.

The ablation study results validate the positive role of the VoxelGrid and ConvLSTM modules in the network, and their impact on detection speed is minimal. This part of the work enables a deeper understanding of the model's working principle, offering valuable guidance on model design and optimization.

## Scalability test

This section studies two contents: Firstly, the performance of event cameras and conventional optical cameras are compared for backlight and dim space object detection. In addition, the influence of aggregation time of event streams on detection performance is studied.

(1) Three classical detection frameworks (Faster R-CNN (*Ren et al., 2017*) , SSD (*Liu et al., 2016*) , YOLOV5 (*Shariff et al., 2022*)) are selected for backlight and dim space object detection, and compared by using frame images and event images. Table 3 shows the detection results on Event_DVS_space7 dataset, among which YOLOV5 has a slightly higher detection accuracy. Compared with the event-based network, the frame-based network has better detection performance, mainly because the frame image resolution is

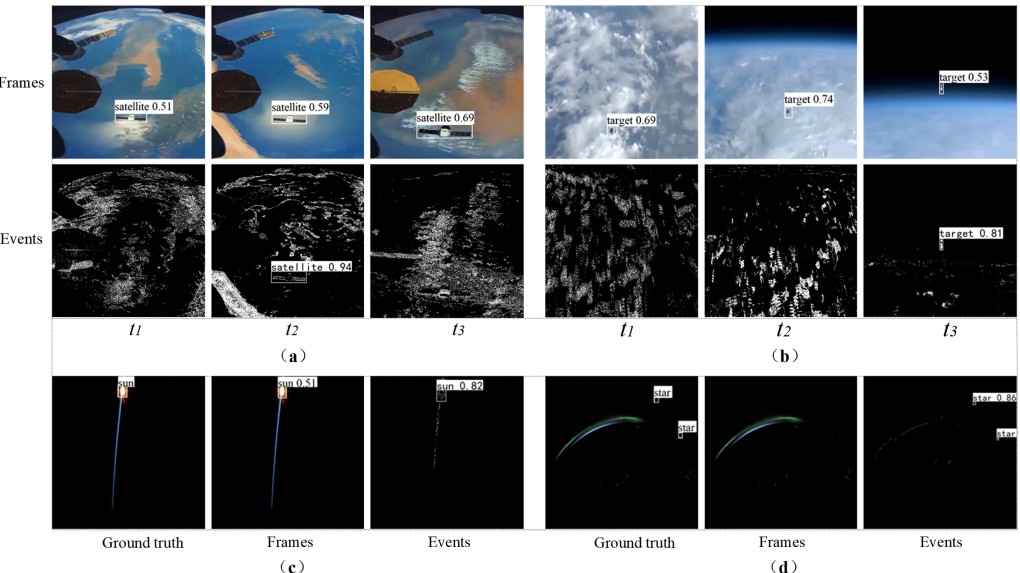

**Figure 10 Visual comparison of detection results between frame images and event images.** (A) Surface object scenes of earth observation. (B) Point object scenes of earth observation. (C) Scenes of backlight. (D) Scenes of dim light. Image source: https://doi.org/10.6084/m9.figshare.26005189.v1.

much higher than the event camera and contains rich RGB information. There are a large number of earth observation, surface target and other scenes in the data set, and the frame-based network has more advantages in the detection of such scenes. But it is difficult to catch the target in backlight and dim light.

The visual detection results of frame images and event images are shown in Fig. 10. Figures 10A and 10B show the surface object and point object scenes of earth observation respectively. The network based on frame images has better detection performance. The main reason is that frame images have higher resolution than event images and contain rich RGB information. Due to the asynchronous characteristics of the event camera, the object may be submerged in the background noise, resulting in missed detection. However, the frame-based detection makes it difficult to capture the object under the backlight and dim light, as shown in Figs. 10C and 10D, which show scenes of backlight and dim light respectively. Event-based detection has obvious advantages in these scenes, especially when the frame image is almost invisible to the tiny object in dim light.

To sum up, frame-based object detection is more suitable for scenes with complex backgrounds, slow or relatively static motion, earth observation, *etc.*, while event-based object detection is more suitable in backlight, dim light, and tiny object scenes. Therefore, the joint detection frameworks based on event image and frame image are more beneficial to the research in the space field. When event streams need static information to complete high-precision identification, frame-based object detection is the main method; However, when the frame image fails to capture objects in challenging scenes, event-based object detection is the main method.

**Table 4 The performance of temporal aggregation sizes.**

| Size (ms) | 25 | 50 | 75 | 100 | 125 | 150 | 200 | 300 |
|---|---|---|---|---|---|---|---|---|
| mAP | 0.457 | 0.493 | 0.563 | 0.641 | 0.643 | 0.646 | 0.652 | 0.655 |
| Runtime (ms) | 36.14 | 42.6 | 61.31 | 70.14 | 130.2 | 188.64 | 276.04 | 384.17 |

It is worth noting that the event-based object detection in this section takes the representation of event images and the YOLOV5 framework as examples. Other event representations and detection frameworks (such as Faster R-CNN, SSD, *etc.*) can be used as alternatives, so this part of the work provides a general interface for the input of asynchronous events.

(2) Temporal aggregation Size: As shown in the Table 4, we investigate the effect of the aggregation time of the input event stream on the final detection performance. The aggregation time was selected as 25, 50, 75, 100, 125, 150, 200 and 300 ms. In the data set, the corresponding mAP differences were increased by 3.6%, 10.6% and 18.4%, respectively, compared with the aggregation time of 25 ms. 18.6%, 19.5% and 19.8%. Obviously, the accuracy tends to continue to improve with the increase of the polymerization size over time. This shows that from the size of the continuous event stream, the longer the aggregation time, the more time information is stored in the spatiotemporal memory module, thus obtaining better detection performance. Therefore, it is reasonable to set the polymerization time to 100 ms for good performance. This precision and speed trade-off is useful in some scenarios where higher precision detection performance is required.

## DISCUSSION

To meet the needs of space object detection under backlight and dim light, this article uses an event camera as the detection load and proposes ACMNet to process the event camera data. The core idea is to first convert the asynchronous event stream into Event Spike Tensor (EST) voxel grid representation through the exponential kernel function, then use a feedforward feature extraction network to extract the spatial features of the object, and introduce ConvLSTM to extract time-series features, and ultimately realize the end-to-end object detection of the asynchronous event stream. This research provides a new possibility for space object detection under backlight and dim light and highlights the advantages of event camera in space object detection.

Compared with models such as SSD and YOLOV5, the accuracy of the proposed algorithm is in the optimal or sub-optimal position for short-distance and surface objects with clearly visible contours to be detected, for distant discrete point objects such as stars, the proposed algorithm is significantly superior to other algorithms, and the results are shown in Table 1. The proposed algorithm is improved according to the benchmark (*Shariff et al., 2022*), and mAP is improved by 12.7%. The reason is that two strategies are introduced: The first is to change the representation of the event streams, that is, to use the EST voxel grid, which allows the asynchronous event streams to retain high-resolution time information. In addition, a ConvLSTM is introduced to extract the time features

between adjacent event streams to improve the detection performance. The contribution of these two modules is further verified in the ablation study, as shown in Table 2, the detection accuracy is improved by 8.6% after the introduction of the ConvLSTM module and 9.2% after the introduction of the Voxel Grid in ACMNet, which is consistent with *Gehrig*'s *et al. (2019)* results. VoxelGrid's characterization method effectively retains the high temporal resolution and spatiotemporal characteristics of the event, especially for the backlight and faint objects lacking obvious spatial features, this characterization retains features of the temporal dimension, which is conducive to object detection. In the scalability test, by comparing the object detection of event cameras and traditional cameras, the results show that the detection accuracy of ACMNet is improved by 3.1% based on YOLOV5 (*Shariff et al., 2022*), which is significantly better than the object detection algorithm based on traditional camera. The reasons are as follows: Firstly, the event camera has the characteristics of high dynamic range and high temporal resolution, and it can maintain high-resolution detection of the object under the backlight and dim light. In addition, the ACMNet algorithm changes the representation of the event streams, allowing high-resolution time-domain features to be retained in the asynchronous event streams; Thirdly, ConvLSTM extracts time-domain features between adjacent event streams to improve the detection performance, and this operation also increases the computational complexity.

Due to the limited dataset for training, the images captured by various sensors are not the same, and are affected by the operating mode of sensors, observation distance, number of objects and other factors. When there is a large difference between the real detection scene and the training dataset scene, higher requirements are put forward for the detection model, which reflects the robustness of the detection algorithm model. Event_DVS_space7 dataset selects the video data recorded in the real space environment which contains a large number of backlight, dim light, and other real space video data. The results of Figs. 8 and 10 show that the algorithm model trained in the Event_DVS_space7 dataset can effectively detect space objects of backlight, dim light, indicating that the Event_DVS_space7 dataset is relatively comprehensive, and the proposed detection algorithm has a high degree of robustness. Therefore, further attention will be paid to the release of the spatial perception dataset to improve the robustness of the proposed detection algorithm and apply it to space object detection tasks under the specific backlight and dim light.

## CONCLUSION

Event cameras exhibit characteristics of high temporal resolution and a broadened dynamic range, offering a novel approach to tackle the application challenges posed by backlight and dim light conditions in space object detection tasks. Therefore, ACMNet is proposed to solve the backlight and dim space object detection problem by using event cameras. EST voxel grid representation and ConvLSTM are introduced to make full use of the temporal information in event streams to better learn spatiotemporal features. The experimental results demonstrate that our ACMNet performs better than the existing event-based object detection methods, shows stronger robustness and adaptability under different operating modes, detection distances, the number of objects, and light intensity.

We believe this research will unlock the potential of event cameras in high-precision space object detection tasks.

The findings of this study are intricately linked to spacecraft's autonomous perception, decision planning, and collision avoidance of space objects. These linkages are analyzed in detail as follows: firstly, the study results verify the effectiveness and advantages of the proposed algorithm for object detection under backlight and dim light conditions. In addition, the applicability of event cameras and traditional optical cameras in the field of space object detection is analyzed in depth. The study results can provide a scientific basis for selecting suitable camera loads in different scenes. Finally, from a practical perspective, this study holds significant value for autonomous perception of hazardous objects and accurate prediction of collision risks. Through the establishment of the detection model, objects in backlight and dim light scenes can be perceived more accurately, thus supporting the relevant departments to develop decision-making planning and avoidance strategies.

Considering the differences in data quality and temporal accuracy between training and actual application data, future work will further adopt transfer learning. The model will be pre-trained on the generated data set, and then fine-tuned on a small amount of real-world event camera data to enhance the generalization and adaptability of the model. Another limitation is that compared to traditional optical cameras, event cameras still have some shortcomings. In subsequent work, we can try to combine traditional optical cameras with event cameras, design a joint spatial object detection architecture based on frames and events, and further develop the application potential of event cameras in the space field.

### Funding
This work was supported by the National Natural Science Foundation of China (No. 12272010). The funders had no role in study design, data collection and analysis, decision to publish, or preparation of the manuscript.

### Grant Disclosures
The following grant information was disclosed by the authors:
National Natural Science Foundation of China: 12272010.

### Competing Interests
Xiaoli Zhou and Chao Bei are employed by CASIC Space Engineering Development Co., Ltd.

### Author Contributions
- Xiaoli Zhou conceived and designed the experiments, performed the experiments, analyzed the data, performed the computation work, prepared figures and/or tables, authored or reviewed drafts of the article, and approved the final draft.
- Chao Bei conceived and designed the experiments, prepared figures and/or tables, authored or reviewed drafts of the article, and approved the final draft.

## Data Availability

The dataset and code are available at figshare: Zhou, Xiaoli (2024). Raw data-code-ACMNet. figshare. Dataset. https://doi.org/10.6084/m9.figshare.26005189.v1.

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
