# Peer review of "Backlight and dim space object detection based on a novel event camera"

_PeerJ Computer Science, doi:10.7717/peerj-cs.2192_

## Round 0.1 · original submission · Minor Revisions

Based on the reviewers' comments, the manuscript could be considered for publication after a revision to address the comments of R3

Reviewer 1 ·

Basic reporting

N/A.

Experimental design

N/A.

Validity of the findings

N/A.

Additional comments

I commend the authors for their extensive data set, compiled over many years of detailed fieldwork. In addition, the manuscript is clearly written in professional, unambiguous language.

Cite this review as

Reviewer 2 ·

Basic reporting

1. According to convention, the first level headings "Conclusions" and "Acknowledgements" should be "Conclusion" and "Acknowledgement".
2. The checkmark in Table 2 is not in the center of the grid.
3. The characters in Figure 1 may be slightly larger.

Experimental design

no comment

Validity of the findings

no comment

Additional comments

no comment

Cite this review as

Reviewer 3 ·

Basic reporting

The paper is generally clear, and the use of professional terminology is appropriate, with a logical structure. However, some sections are verbose and repetitive, such as parts of the introduction, which could be more concise.The paper mentions the construction and use of the Event_DVS_space7 dataset but does not provide detailed instructions on how to access this data. It is recommended to include a method for accessing the dataset in the appendix to facilitate replication by other researchers.

Experimental design

The research is innovative in applying event cameras to space object detection, aligning well with the journal's scope. The proposed ACMNet network effectively addresses the problem of detecting space objects under backlight and dim conditions, filling a gap in existing knowledge. The methodology is described in detail, but some steps could be more explicit.

Validity of the findings

The paper demonstrates the effectiveness of ACMNet in specific conditions but lacks a thorough discussion of its impact and novelty. It is suggested to include more discussion on potential impacts and possible improvements in other application scenarios to provide a comprehensive evaluation of its innovativeness. The results section clearly demonstrates the advantages of ACMNet. However, the statistical analysis of experimental results could be more detailed.The conclusions summarize the main findings and relate closely to the research question. They are clear and supported by the results, but it is suggested to discuss the limitations of the study and future research directions in the conclusion to provide a more comprehensive perspective.

Cite this review as

---

## Round 0.2 · accepted · Accept

Dear authors,

Please read carefully at the reviewers' comments and make the revisions accordingly, if there is any.

Thank you for your contribution,
Best regards.

Reviewer 2 ·

Basic reporting

Some English sentences in certain areas can be polished.

Experimental design

no comment

Validity of the findings

no comment

Additional comments

no comment

Cite this review as

Reviewer 3 ·

Basic reporting

no comment

Experimental design

no comment

Validity of the findings

no comment

Additional comments

no comment

Cite this review as